# Bacteriophage-Associated Antimicrobial Resistance Genes in Avian Pathogenic *Escherichia coli* Isolated from Brazilian Poultry

**DOI:** 10.3390/v15071485

**Published:** 2023-06-30

**Authors:** Giulia Von Tönnemann Pilati, Rafael Dorighello Cadamuro, Vilmar Benetti Filho, Mariane Dahmer, Mariana Alves Elois, Beatriz Pereira Savi, Gleidson Biasi Carvalho Salles, Eduardo Correa Muniz, Gislaine Fongaro

**Affiliations:** 1Laboratory of Applied Virology, Department of Microbiology, Immunology and Parasitology, Federal University of Santa Catarina, Florianópolis 88040-900, Brazil; cadamuro.rafael@gmail.com (R.D.C.); vilmarbf98@gmail.com (V.B.F.); maariana.eloiss@gmail.com (M.A.E.); beasavis2@gmail.com (B.P.S.); gleidson.salles@zoetis.com (G.B.C.S.); 2Zoetis Industry of Veterinary Products LTDA, São Paulo 04709-111, Brazil; eduardo.muniz@zoetis.com

**Keywords:** multi-resistance, colibacillosis, phages, horizontal transfer genes (HGT), whole genome sequencing (WGS)

## Abstract

Colibacillosis is a disease caused by *Escherichia coli* and remains a major concern in poultry production, as it leads to significant economic losses due to carcass condemnation and clinical symptoms. The development of antimicrobial resistance is a growing problem of worldwide concern. Lysogenic bacteriophages are effective vectors for acquiring and disseminating antibiotic resistance genes (ARGs). The aim of this study was to investigate the complete genome of *Escherichia coli* isolates from the femurs of Brazilian broiler chickens in order to investigate the presence of antimicrobial resistance genes associated with bacteriophages. Samples were collected between August and November 2021 from broiler batches from six Brazilian states. Through whole genome sequencing (WGS), data obtained were analyzed for the presence of antimicrobial resistance genes. Antimicrobial resistance genes against the aminoglycosides class were detected in 79.36% of the isolates; 74.6% had predicted sulfonamides resistance genes, 63.49% had predicted resistance genes against β-lactams, and 49.2% of the isolates had at least one of the tetracycline resistance genes. Among the detected genes, 27 have been described in previous studies and associated with bacteriophages. The findings of this study highlight the role of bacteriophages in the dissemination of ARGs in the poultry industry.

## 1. Introduction

In 2022, chicken meat production worldwide was 101.086 million tons. Brazil is the second-largest producer of chicken meat in the world, producing 14.524 million tons in 2022 and exporting 33.2% of its production, which also makes it the largest exporter in the world [1]. This significant increase is due to the rapidly growing poultry industry and is related to factors such as genetic selection and the implementation of various measures aimed at improving the health and performance of the birds [2].

Avian colibacillosis, a naturally opportunistic disease with local or systemic manifestation, is caused by avian pathogenic *Escherichia coli* (APEC). It is important to note that not every case of colibacillosis is necessarily caused by APEC [3,4]. The disease is distributed worldwide and has a major economic impact, especially in the broiler industry due to mortality, morbidity, lack of uniformity in a flock, lowered production, and increased condemnation at the slaughter [5].

The agent is responsible for causing different clinical conditions in birds, such as airsacculitis, cellulitis, coligranuloma, colisepticemia, pericarditis, peritonitis, pleuropneumonia, pneumonia, omphalitis, salpingitis, swollen head syndrome (SCI), panophthalmos, osteomyelitis, oophoritis, and synovitis [5,6,7].

The most commonly used antimicrobials in the treatment of avian colibacillosis are β-lactams (penicillins, cephalosporins), fluorquinolones, lincosamides, macrolides, quinolones, sulfonamides, and tetracyclines [5,8,9,10]. Currently, many of the antimicrobials used in poultry production are also used in human medicine. This has raised concerns about the potential transfer of antibiotic resistance genes between animals and humans [11].

Apart from their use for the treatment and prophylaxis of human and animal infections, antibiotics are widely used as metaphylactic agents and growth promoters in animal production. Such practices, however, increase selective pressure and may favor the development of antimicrobial resistance [12,13,14].

The development of antimicrobial resistance is a complex process. Resistance can be classified as either inherent or acquired. Inherent resistance is the natural ability of some bacteria to resist certain antibiotics due to intrinsic properties such as their cell wall structure or metabolic pathways. Acquired resistance, on the other hand, is the result of genetic changes in bacteria, such as mutations or the transfer of resistance genes from other bacteria [9,15,16,17,18]. This transfer can occur through several mechanisms, including horizontal gene transfer (HGT), which involves the transfer of genetic material between different bacterial cells. HGT can occur through three well-known mechanisms: conjugation, transformation, and transduction, which allow for the transfer of genetic material, such as plasmids or bacteriophages. Some of these mechanisms are facilitated by mobile genetic elements (MGEs), which include phages, plasmids, genomic islands, and integrative conjugative elements (ICEs) [19].

Bacteriophages, also known as phages, are viruses that infect and replicate within bacteria. They are the most abundant biological entities on Earth and play a crucial role in shaping microbial communities and the evolution of bacterial populations. However, recent studies have highlighted the potential of phages to transfer and propagate antibiotic resistance genes, a phenomenon that can have significant consequences for human health [20,21].

Phages can infect bacteria as lytic phages or as lysogenic phages. Lytic phages replicate within the host cell, leading to its destruction and the release of progeny phages. In contrast, lysogenic phages integrate their DNA into the host genome and can remain dormant for long periods of time without causing lysis. During this time, they can replicate along with the host chromosome and be transmitted vertically to daughter cells [22,23].

Although bacteriophages have been explored as potential biocontrol agents against bacterial infections, recent studies have raised concerns about their unintended impact on bacterial populations. One such concern is the potential for phages to inadvertently transfer genetic material, including antibiotic resistance genes, between bacteria [24,25,26].

Phages have been shown to carry genes encoding for antibiotic resistance and to transfer them to susceptible bacteria through a process called lysogenic conversion. This process occurs when a lysogenic phage integrates its DNA into the host chromosome, carrying with it a gene for antibiotic resistance. The resistance gene can then be expressed by the host bacterium and transferred horizontally to other bacteria through mobile genetic elements, such as plasmids or transposons [22,27,28].

The aim of this study was to investigate the complete genome of avian pathogenic *Escherichia coli* (APEC) isolates from the femurs of Brazilian broilers (*Gallus gallus domesticus*) in order to investigate the presence of antimicrobial resistance genes associated with bacteriophages.

## 2. Materials and Methods

### 2.1. Sample Collection

This study evaluated 100 batches of chicken (*Gallus gallus domesticus*) carcasses collected in Brazil between August and December 2021 for *E. coli* research. In this case, the femurs were chosen because they were intact and were more likely to contain highly pathogenic isolates. The batches were selected based on a history of respiratory problems, and some showed clinical respiratory signs, such as sneezing, snoring, and nasal discharge. The femur samples were collected from six different states in Brazil: Paraná (*n* = 30), Santa Catarina (*n* = 15), Rio Grande do Sul (*n* = 15), São Paulo (*n* = 10), Minas Gerais (*n* = 10), and Ceará (*n* = 20), which represent approximately 80% of chicken meat production in Brazil (ABPA, 2023). For each batch, three femurs, three livers, and three spleens were collected and processed at the Laboratory of Applied Virology (LVA) of the Federal University of Santa Catarina (UFSC).

### 2.2. Escherichia coli Isolation

To isolate *E. coli*, femurs were processed aseptically, and a swab was used to collect the bone marrow, which was suspended in a saline buffer. The swabs were then inoculated into MacConkey agar and incubated at 37 °C for 24 h. After incubation, typical *E. coli* colonies (indicated by a pink staining) were picked and isolated. These isolates were then stored at −80 °C for further analysis. Therefore, each isolate obtained corresponds to a batch of birds.

### 2.3. Avian Pathogenic Escherichia coli Molecular Confirmation

To confirm the presence of APEC in the femur isolates, conventional PCR reactions were performed. Genomic DNA was extracted using the phenol–chloroform method that was adapted for this study. Colonies were grown in LB broth for 6 h at 37 °C and then cultured on MacConkey agar and incubated for 24 h at 37 °C. One colony was selected and inoculated again in LB broth for 18 h. The samples were centrifuged twice for 10 min at 700× *g*, the supernatant was discarded, and the pellet was washed twice in PBS pH 7.2. For sample lysis, the pellet was suspended in 200 μL of lysis buffer containing 26.2 μL of Proteinase K and incubated at 56 °C for 30 min. For DNA extraction, an equal volume of balanced phenol was added, homogenized by inversion, and centrifuged for 10 min at 14,000× *g* at room temperature. The aqueous upper phase was transferred to a new microtube, and the same volume of a phenol–chloroform solution (1:1) was added, homogenized by inversion, and centrifuged for 10 min at 14,000 at 4 °C. The aqueous upper phase was transferred to a new microtube, and an equal volume of chloroform was added, homogenized via inversion, and centrifuged for 10 min at 14,000× *g* at 4 °C. The aqueous upper phase was transferred to a new microtube, and it was added to 2.5× the total volume of ice-cold 100% ethanol. For DNA precipitation, the microtube was kept at −20 °C for 1 h and then centrifuged for 30 min at 14,000× *g* at 4 °C. The supernatant was discarded, and the pellet was washed twice with 500 μL of ice-cold 70% ethanol and centrifuged for 10 min at 14,000× *g* at 4 °C. The pellets were dried at 37 °C and suspended in 50 μL of type I ultrapure water. DNA was quantified by optical density in spectrophotometry using the NanoVue device and stored at −20 °C.

For the conventional polymerase chain reaction (PCR), the genes (*iroN*, *ompT*, *hlyF*, *iss*, and *iutA*) described in Table 1 were used as the minimal virulence predictors of APEC [29]. Amounts of 2 mM magnesium chloride, 0.25 mM deoxyribonucleotide phosphates, 0.3 μM of each primer, 1 U of Taq DNA polymerase GoTaq^®^ DNA Polymerase (Promega, Madison, WI, USA), and buffer 1× Green GoTaq^®^ Reaction Buffer (Promega, Madison, WI, USA) were mixed with 3 μL of sample and sterile ultrapure water to make 25 μL. The reactions were carried out in a thermocycler using the following cycling parameters: 94 °C for 2 min, 35 cycles of 94 °C for 30 s, 63 °C for 30 s, 68 °C for 3 min, and a final cycle of 72 °C for 10 min.

The samples were subjected to 1% horizontal agarose gel electrophoresis, using GelRed as a DNA intercalating agent. Amplicon sizes were determined by comparison with the low molecular weight (LMW) molecular weight marker.

### 2.4. Species Confirmation by Sequencing

Cultivation was carried out on MacConkey agar, and characteristic colonies of *E. coli* were used in the complete genome sequencing; for this, an isolated colony was used.

To prepare the genomic DNA libraries for sequencing, the Illumina DNA Prep—Nextera kit was used, and the libraries were quantified with the Collibri Library Quantification Kit following the manufacturer′s instructions. The Nextseq System was used for paired-end sequencing, generating raw reads based on 300 cycles with 2 × 150 bp reads.

After obtaining the raw data from the MiSeq platform, Phred quality score was used to process the data, and reads with Q scores less than 20 were removed. Adapters and low-quality segments of sequences were also discarded during quality control (QC).

Genome assembly was performed using oneshotWGS v1.9, which includes A5 assembly and additional steps for adapter trimming, quality filtering, and error correction to generate scaffolds [31]. Chimeric segments were removed to obtain the best assembly, and assembly statistics were generated using QUAST 5.2.0 software [32].

Genome annotation was carried out using Prokka 1.14.6 software and customized proprietary databases containing curated gene sequences from public databases [33].

The genome sequencing and assembly data were deposited in the NCBI database with the Bioproject accession number (PRJNA917297).

In silico species confirmation was performed using the neogSpecies module, which used an ANI analysis with a cutoff of 97% to estimate genome species [34].

### 2.5. Detection of Antimicrobial Resistance Genes

The presence of antimicrobial resistance genes was evaluated with the complete genome sequencing data of the samples using the program Abricate 1.0.1 (https://github.com/tseemann/abricate (accessed on 24 May 2023), with the Resfinder database version (using minimum coverage and identity of 80% [35].

## 3. Results

### 3.1. E. coli and APEC Confirmation

A total of 63 characteristic *Escherichia coli* isolates were obtained from femurs. All isolates were confirmed as *Escherichia coli* via sequencing. Of the 63 isolates, 58 (92%) had between 3 and 5 of the genes considered as minimum predictors and could be characterized as avian pathogenic *Escherichia coli* (APEC). Of these, 40 (63.4%) showed the 5 genes, 14 (22.2%) presented 4 genes, 4 (6.3%) showed 3 genes, and 4 (6.3%) showed between 1 gene and 2 genes.

### 3.2. Detection of Antimicrobial Resistance Genes

All isolates examined exhibited the presence of at least one antimicrobial resistance gene (ARG) (Figure 1). The resistance genes *sul2* (57.8%), *ant (3″)-Ia* (51.5%), *qacE* (50.79%), *sul1* (52.38%), *aph (6)-Id* (32.8%), *aac (3)-VIa* (35.9%), *tet (A)* (31.2%), *tet (B)* (21.8%), *floR* (20.3%), *aadA2* (20.3%), and *aph (3″)-Ib* (20.3%) were identified in 20% or more of the analyzed isolates.

Among the resistance genes detected, those that were present in more than 20% of the samples from the south region, represented by the states of Paraná, Santa Catarina and Rio Grande do Sul, were *aac (3)-VIa* (35.29%), *aadA2* (26.47%), *ant (3″)-Ia* (50%), *aph (6)-Id* (29.41%), *blaCTX-M-2* (20.58), *sul1* (47.05%), *sul2* (47.05%), and *tet (A)* (29.41%).

In the southeast region, represented by the states of Minas Gerais and São Paulo, the most prevalent genes among batches are *aac (3)-Vla* (36.84%), *ant (3″)-Ia* (52.63%), *aph (3″)-Ib* (21.05%), *aph (6)-Id* (31.57%), *sul1* (63.15%), *sul2* (73.68%), *tet (A)* (31.57%), and *tet (B)* (26.31%).

In the northeast region, represented by the state of Ceará, the resistance genes detected in more than 20% of the samples were *aac (3)-Via* (40%), *aadA2* (20%), *ant (3″)-Ia* (50%), *aph (3″)-Ib* (40%), *aph (3′)-Ia* (20%), *aph (6)-Id* (40%), *blaTEM-1* (20%), *blaTEM-1B* (20%), *dfrA12* (20%), *floR* (40%), *qnrB19* (20%), *sul1* (50%), *sul2* (80%), *tet (A)* (40%), and *tet (B)* (30%).

Figure 2 relates the co-occurrence of resistance genes associated with two different classes of antimicrobials. The image presents the combinations of classes that appear in 5% or more of the batches used in the study. It was observed that 37 (58.73%) of the evaluated bird flocks exhibited a co-occurrence of resistance genes to aminoglycosides and beta-lactams. Additionally, 44 (69.84%) flocks demonstrated a co-occurrence of genes conferring resistance to aminoglycosides and sulfonamides. Furthermore, concerning beta-lactams, 33 batches showed genes of resistance to this class along with genes of resistance to sulfonamides. Additionally, in 26 (41.26%) of the batches, genes of resistance to sulfonamides and tetracyclines were detected.

Among the detected genes, 27 have been described in previous studies and associated with phages: *aac (3)-VIa*, *aph (3′)-Ia*, *blaCTX-M-1*, *blaCTX-M-15*, *blaCTX-M-164*, *blaCTX-M-2*, *blaCTX-M-55*, *blaCTX-M-8*, *blaSHV-12*, *blaSHV-187, blaTEM-106*, *blaTEM-14*, *dfrA12*, *dfrA15*, *catA1*, *floR*, *mph (A)*, *qnrS1*, *sul1*, *tet (A)*, *tet (B)*, and *qacE*.

## 4. Discussion

Antimicrobial resistance is a crucial public health issue, and its emergence and spread are the results of a complex interplay of multiple interconnected factors. Due to the emergence of antibiotic-resistant bacteria outpacing the development of new antibiotics, there are currently only a limited number of antibiotics available without resistance [36].

Studies developed in the last years in different countries, including Australia, Jordan, Pakistan, Italy, and Thailand identified several resistance genes in avian pathogenic *Escherichia coli* isolates from different organs. The genes described were *aadA1*, *aadA2*, *aac (3)lid*, *aaac (3)-ant (3″)-Ia*, *blaTEM-1A/B/C*, *blaCTX-M-15*, *bla-SHV*, *cat1*, *cmlA1*, *dfrA5*, *dfrA1*, *floR*, *dfrA1*, *dfrA5*, *dfrA12*, *dfrA14*, *dfrA17*, *sat2*, *strA*, *strB*, *sul1*, *sul2*, *sul3*, *tet (A)*, *tet (B)*, and *tet (C)* [9,14,37,38,39]. This fact highlights the growing concern about antimicrobial resistance and the consequences related to this problem.

ResFinder is a web-based method in which the identification of antimicrobial resistance genes acquired in whole-genome data is performed through BLAST [35]. A study developed in 2014 used ResFinder to identify acquired antibiotic resistance genes associated with *E. coli* in genomes of phages collected from public databases and of prophages predicted from bacterial genomes. The gene *catA1,* which belongs to the class of phenols, was found in the phage genome. In the current study, 3.125% of *E. coli* isolates harbored the gene. In the prophage set, 14 predicted prophages were found to contain a total of 31 resistance genes. The genes identified in prophages were *catA1*, *qacE*, *aaA5*, *mph (A)*, *sul1*, *dfrA17*, *blaTEM-1*, *qacE1*, and *aph (3′)-Ia* [40]. In the current study, *catA1* was detected in 3.17% of the isolates—*blaTEM-1* (22.22%), *mph (A)* (1.58%), *qacE* (50.79%), and *sul1* (52.38%).

The present study described the presence of the genes *dfrA12*, *dfrA15*, *floR*, *qnrA*, and *qnrS*. Recent research demonstrated that bacteriophages may serve as a reservoir for these resistance genes [41,42,43,44,45,46].

β-lactams are a group of antibiotics that include penicillins, cephalosporins, carbapenems, and monobactams. These drugs are widely used in veterinary medicine to treat bacterial infections in animals [47]. In this study, predicted resistance genes against β-lactams were found in 63.49% of the isolates harboring one or more of the following genes: *blaCTX-M-1*, *blaCTX-M-15*, *blaCTX-M-164*, *blaCTX-M-2*, *blaCTX-M-55*, *blaCTX-M-8*, *blaSHV-12*, *blaSHV-187*, *blaTEM-106*, *blaTEM-141*, *blaTEM-1A*, and *blaTEM-1B*. Some studies analyzed the genomes of phages obtained from different matrices—such as sewage, wastewater treatment stations, poultry meat, phages found in the environment, products, pig feces, meat, pork, beef and chicken minced meat, ham and mortadella, chicken feces, chicken liver, and compost—and have described the presence of all genes except *blaCMY-2* [41,42,43,44,45,48,49,50,51,52].

Tetracyclines are a class of antibiotics commonly used in veterinary medicine to treat a variety of bacterial infections in animals. They are effective against a wide range of bacteria, including both gram-positive and gram-negative bacteria, and are commonly used to treat respiratory, urinary tract, and skin infections in animals [47,53]. In the current study, 49.2% of the isolates harbored at least one tetracycline resistance gene of *tet (A)*, *tet (B)*, and *tet (D)*. Some studies analyzed the genome of phages obtained from chicken meat products described the presence of *tet (A)* [42,49].

Aminoglycosides are commonly used in veterinary medicine to treat bacterial infections in various animal species, including poultry. These antibiotics are effective against a wide range of gram-negative bacteria, including *Escherichia coli*, *Salmonella*, and *Pseudomonas aeruginosa*. In the poultry industry, aminoglycosides are frequently used to prevent and treat respiratory and enteric infections caused by gram-negative bacteria [54]. One of the most commonly used aminoglycosides in veterinary medicine is gentamicin. In the present study, 78.1% of the APEC isolates harbored one or more aminoglycoside resistance genes (*aac (3)-IId*, *aac (3)-IVa*, *aac (3)-VIa*, *aadA12*, *aadA2*, *aadA5*, *ant (2″)-Ia*, *ant (3″)-Ia*, *aph (3′)-Ia*, *aph (3″)-Ib*, *aph (4)-Ia*, and *aph (6)-Idii)*. Studies analyzed the genome of phages isolated from poultry meat derivatives, pig feces, chicken feces and chicken liver and described the presence of *aac (6′)-Im*, *aac-(Ib)-cr*, *aac (3)-Via*, *aphA1*, *aadA2*, *aph (3′)-IIIa*, *ant (6)-Ia*, *aph (2″)-Ig*, and *aph (3′)-III* resistance genes [42,43,45,46,50,51].

Sulfonamides are a class of antibiotics widely used in veterinary medicine, including in the poultry industry. Sulfonamides are effective against a wide range of gram-positive and gram-negative bacteria and are commonly used to treat respiratory, enteric, and urinary tract infections in poultry [47]. The current study shows that 74.6% of the isolates carried predicted resistance genes against sulfonamides (*sul1*, *sul2*, and *sul3*). Previous studies described the presence of sulfonamide resistance genes in bacteriophage isolated from pig feces, meat, pork, beef, chicken minced meat, ham and mortadella, dog urinary tract, chicken feces, chicken liver, and compost [43,44,45,50,51,52,55].

These genes have been detected in the genome of Enterobacteria phage P7 (accession number AF503408) [40]; phages from the *Myoviridae*, *Siphoviridae*, *Podoviridae*, and *Inoviridae* families [48,49,50]; and *Drexlerviridae* [51]. The gene *qacE* was identified in the prophage (uid33411_0.8) of *Escherichia coli* IAI39; the genes *catA1*, *blaTEM-1*, *mph (A)*, and *sul1;* and in prophage (uid33415_0.1) of *Escherichia coli* UMN026 [40].

## 5. Conclusions

The confirmation of ARGs in association with bacteriophages is still lacking, primarily due to the absence of specific bacteriophage isolations and sequencing. Without these essential investigations, a direct correlation between bacteriophages and ARGs cannot be established. Consequently, to obtain a comprehensive understanding of their relationship and to develop effective strategies against antibiotic resistance, it is imperative to conduct thorough studies that isolate and sequence bacteriophages.

In line with this objective, the findings of our study shed light on the prevalence of antimicrobial resistance genes in *E. coli* isolates obtained from broilers in Brazil. These results highlight the possibly significant role played by phages in the dissemination of these ARGs within the poultry industry. This is an important health alert in order to expand antimicrobial studies, considering their interaction in bacteria infected with phages and the host–parasite relationship in health studies and combating bacterial infection since not enough is known about phage–bacteria interaction in antimicrobial efficiency studies.

## Figures and Tables

**Figure 1 viruses-15-01485-f001:**
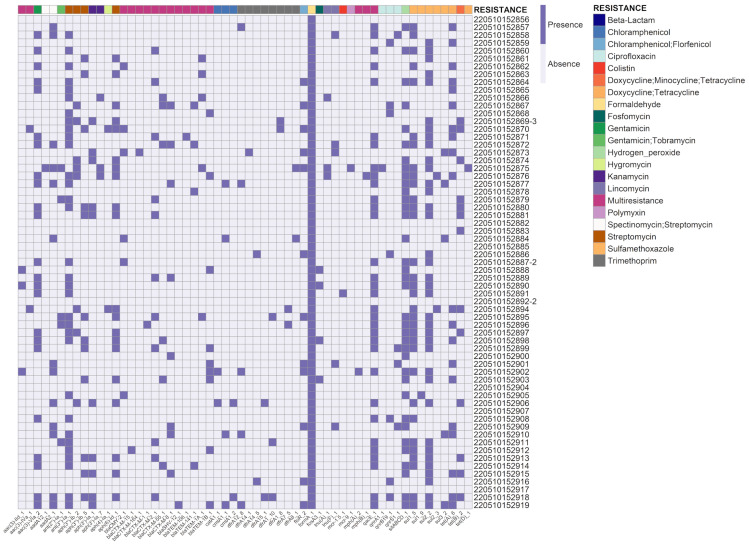
Presence or absence of antibiotic resistance genes was assessed in *E. coli* isolates submitted for sequencing. Each line in the dataset corresponds to a unique sample, while the columns represent the identified resistance genes and the corresponding antibiotics or antibiotic classes.

**Figure 2 viruses-15-01485-f002:**
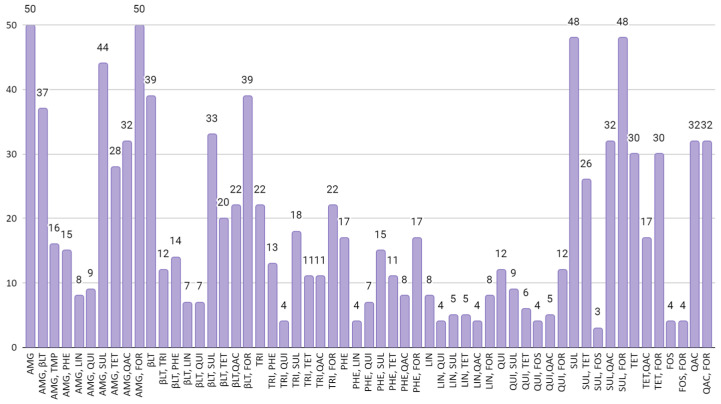
Co-occurrence of resistance genes associated with two different classes of antimicrobials, including AMG (aminoglycosides), βLT (β -lactams), TMP (trimethoprim), PHE (phenicol), LIN (lincosamides, QUI (quinolones), SUL (sulfonamides), TET (tetracyclines), QAC (quaternary ammonium compound), FOR (formaldehyde), and FOS (fosfomycin). The image presents the combinations of classes that appear in 5% or more of the batches in the study.

**Table 1 viruses-15-01485-t001:** Target gene, sequence of primers, amplicon size, and reference.

Target Gene	Primer Sequence	Amplicon Size	Reference
*iroN*	5′-AAGTCAAAGCAGGGGTTGCCCG-3′	667 bp	[30]
	5′-GATCGCCGACATTAAGACGCAG-3′		
*ompT*	5′-TCATCCCGGAAGCCTCCCTCACTACTAT-3′	496 bp	[29]
	5′-TAGCGTTTGCTGCACTGGCTTCTGATAC-3′		
*hlyF*	5′-GGCCACAGTCGTTTAGGGTGCTTACC-3′	450 bp	[29]
	5′-GGCGGTTTAGGCATTCCGATACTCA-3′		
*iss*	5′-CAGCAACCCGAACCACTTGATG-3′	323 bp	[30]
	5′-AGCATTGCCAGAGCGGCAGAA-3′		
*iutA*	5′-GGCTGGACATCATGGGAACTGG-3′	302 bp	[29]
	5′-CGTCGGGAACGGGTAGAATCG-3′		

## Data Availability

The genome sequencing and assembly data were deposited in the NCBI database with the Bioproject accession number (PRJNA917297).

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
