# Peer review of "Bacteriophage-Associated Antimicrobial Resistance Genes in Avian Pathogenic Escherichia coli Isolated from Brazilian Poultry"

_viruses, 2023, doi:10.3390/v15071485_

Round 1

Reviewer 1 Report

Dear Authors,

the topic covered is very interesting and current. Within this field there are not many published articles on the investigation of the complete genome of avian pathogen Escherichia coli (APEC) isolates in order to investigate the presence of bacteriophage-associated antimicrobial resistance genes. Furthermore, the sampling method which also includes the femurs of broiler chickens (Gallus gallus domesticus) taken from different lots from different Brazilian regions, makes the experimental design of the study very accurate and reliable in the results.

I am sure that the data emerging from this study will be useful for enriching the scientific literature and will be able to promote the activation of the sensitivity of readers and scholars on this issue.

There are only a few suggestions I would like to make:

- modify the keywords, without repeating those present in the Title of the paper;

- add possible limits that the study presents;

- integrate the concluding paragraph or add the conclusions section (although not mandatory).

Author Response

Response to Reviewer 1 Comments:

Point 1:  modify the keywords, without repeating those present in the Title of the paper;

Response 1: Thank you for the suggestion, keywords were change: Multi-resistance; colibacillosis; Phages; horizontal transfer genes (HGT); whole genome sequencing (WGS)

Point 2:  add possible limits that the study presents;

Response 2: Thank you for the suggestion, in the conclusion section, we added a paragraph demonstrating the limitation of the work, which uses bacterial DNA to correlate resistance genes that have already been described in phages in previous works:

“The confirmation of ARGs in association with bacteriophages is still lacking, primarily due to the absence of specific bacteriophage isolations and sequencing. Without these essential investigations, a direct correlation between bacteriophages and ARGs cannot be established. Consequently, to obtain a comprehensive understanding of their relationship and to develop effective strategies against antibiotic resistance, it is imperative to conduct thorough studies that isolate and sequence bacteriophages.

In line with this objective, the findings of our study shed light on the prevalence of antimicrobial resistance genes in E. coli isolates obtained from broilers in Brazil. These results highlight the possibly significant role played by phages in the dissemination of these ARGs within the poultry industry. This discovery serves as a crucial health alert, emphasizing the need to expand antimicrobial research efforts, particularly in the context of phage-bacteria interactions and the host-parasite relationship in health studies.”

Point 3: integrate the concluding paragraph or add the conclusions section (although not mandatory)

Response 3: Thank you for the observation. The conclusion was added with appropriate changes:
4. Conclusion

in prophage (uid33415_0.1) of Escherichia coli UMN026 [40].

5. Conclusion

The confirmation of ARGs in association with bacteriophages is still lacking, primarily due to the absence of specific bacteriophage isolations and sequencing. Without these essential investigations, a direct correlation between bacteriophages and ARGs cannot be established. Consequently, to obtain a comprehensive understanding of their relationship and to develop effective strategies against antibiotic resistance, it is imperative to conduct thorough studies that isolate and sequence bacteriophages.

The findings of our study shed light on the prevalence of antimicrobial resistance genes in E. coli isolates obtained from broilers in Brazil. These results highlight the possibly significant role played by phages in the dissemination of these ARGs within the poultry industry. This is an important health alert in order to expand antimicrobial studies, considering their interaction in bacteria infected with phages and the host-parasite relationship in health studies and combating bacterial infection since not enough is known about phage-bacteria interaction in antimicrobial efficiency studies.”

Reviewer 2 Report

In this multiauthor manuscript, the authors have shown that bacteria in the Brazilian poultry can carry several antibiotic-resistance genes in their prophage sequences. Lysogenic phages are long known to carry a variety of non-phage genetic materials besides resistance, such as toxins, transposable genetic elements, conversion genes that alter the genotype and phenotype of the resident bacteria. Nevertheless, this report in a cohort chickens of a major meat-producing nation is of importance for both economical and public health reasons. The studies are generally well done, combining field collection, isolation, as well as modern sequence analysis. What the authors have found is not unexpected, i.e., the samples contain resistance genes against the same antibiotics that were used in the poultry and perhaps elsewhere in the environment that the birds were exposed to. Essentially all major antibiotics were targeted, raising concern.

I do have some queries and suggestions, as described below.

1. Figure 1, which is the only Result figure, appears to show multiple resistance genes in a single sample. It is very important to clarify this. Does it mean that a "single" prophage DNA contains multiple resistance genes, or, each sequencing sample contained a "mixture" of DNA? Basically, the term "samples" should be better described. The difference is important since two resistance markers in a single DNA would indicate recombination, following horizontal transmission.

Perhaps the best way, if not the only way, this can be distinguished is to grow the bacteria from each femur sample on appropriate nutrient agar plates (e.g., LB, tryptone, or blood agar) to clearly isolated single colonies and then purify the DNA from each colony and subject to sequencing. That could lead to interesting results, as I am sure the authors understand.

2. Only limited analysis of the Fig. 1 data was done, just showing percentage of each resistance in different poultry populations. The authors can add another meta-analysis that would be interesting to the readers: the co-occurrence of two different antibiotic resistance genes. For example, do beta-lactam resistance genes have a greater tendency to occur together with tet-resistance ones, or more with aminoglycoside resistance. If yes, it may correlate with the farmer's practice of which two antibiotics they use together, i.e. do they use 'penicillin' and 'tetracycline' classes together more often than 'pencillin' and 'streptomycin' classes together? If that answer is yes, then Comment 1 will be relevant to investigate, i.e., whether pencillin and tetracycline resistance sequences occur in the same prophage DNA.

Author Response

Response to Reviewer 2 Comments

Point 1: Figure 1, which is the only Result figure, appears to show multiple resistance genes in a single sample. It is very important to clarify this. Does it mean that a "single" prophage DNA contains multiple resistance genes, or, each sequencing sample contained a "mixture" of DNA? Basically, the term "samples" should be better described. The difference is important since two resistance markers in a single DNA would indicate recombination, following horizontal transmission.

Response 1: Thank you for the observation.

The information was adedded in Methodology (2.2): “Therefore, each isolate obtained corresponds to a batch of birds.”

Perhaps the best way, if not the only way, this can be distinguished is to grow the bacteria from each femur sample on appropriate nutrient agar plates (e.g., LB, tryptone, or blood agar) to clearly isolated single colonies and then purify the DNA from each colony and subject to sequencing. That could lead to interesting results, as I am sure the authors understand.

Response 1:  Thank you for the observation.

The information was adedded in Methodology (2.4):

Cultivation was carried out on MacConkey agar, and characteristic colonies of E. coli were used in the complete genome sequencing, an isolated colony was used.

Point 2: Only limited analysis of the Fig. 1 data was done, just showing percentage of each resistance in different poultry populations. The authors can add another meta-analysis that would be interesting to the readers: the co-occurrence of two different antibiotic resistance genes. For example, do beta-lactam resistance genes have a greater tendency to occur together with tet-resistance ones, or more with aminoglycoside resistance. If yes, it may correlate with the farmer's practice of which two antibiotics they use together, i.e. do they use 'penicillin' and 'tetracycline' classes together more often than 'pencillin' and 'streptomycin' classes together? If that answer is yes, then Comment 1 will be relevant to investigate, i.e., whether pencillin and tetracycline resistance sequences occur in the same prophage DNA.

Response 2:

 We appreciate the reviewer’s contribution. We performed the analysis and correlated the presence of two classes according to the batches. The results are shown in the image, where we selected the co-occurrences that appear in more than 5% of the batches.

“Figure 2 relates the co-occurrence of resistance genes associated with two different classes of antimicrobials. The image presents the combinations of classes that appear in 5% or more of the batches used in the study. It was observed that 37 (58.73%) of the evaluated bird flocks exhibited a co-occurrence of resistance genes to aminoglycosides and beta-lactams. Additionally, 44 (69.84%) flocks demonstrated a co-occurrence of genes conferring resistance to aminoglycosides and sulfonamides. Furthermore, concerning beta-lactams, 33 batches showed genes of resistance to this class along with genes of resistance to sulfonamides. And, in 26 (41.26%) of the batches, genes of resistance to sulfonamides and tetracyclines were detected.”

“Figure 2. Co-occurrence of resistance genes associated with two different classes of antimicrobials, where AMG (Aminoglycosides), βLT (β -lactams), TMP (Trimethoprim), PHE (Phenicol), LIN (Lincosamides, QUI (Quinolones), SUL (Sulfonamides), TET (Tetracyclines), QAC (Quaternary Ammonium Compound), FOR (Formaldehyde), FOS (Fosfomycin). The image presents the combinations of classes that appear in 5% or more of the batches in the study. “